# Synthesis and assembly of extended quintulene

Hao Hou[1,4], Xin-Jing Zhao[1,4], Chun Tang[1,4], Yang-Yang Ju[1], Ze-Ying Deng[1], Xin-Rong Wang[1], Liu-Bin Feng[1], Dong-Hai Lin[1], Xu Hou [1], Akimitsu Narita [2], Klaus Müllen[2,3] & Yuan-Zhi Tan [1✉]

Quintulene, a non-graphitic cycloarene with fivefold symmetry, has remained synthetically elusive due to its high molecular strain originating from its curved structure. Here we report the construction of extended quintulene, which was unambiguously characterized by mass and NMR spectroscopy. The extended quintulene represents a naturally curved nanocarbon based on its conical molecular geometry. It undergoes dimerization in solution via $\pi-\pi$ stacking to form a metastable, but isolable bilayer complex. Thermodynamic and kinetic characterization reveals the dimerization process as entropy-driven and following second-order kinetics with a high activation energy. These findings provide a deeper understanding of the assembly of conical nanocarbons. Comparison of optical properties of monomer and dimer points toward a H-type interlayer coupling in the dimer.

[1] State Key Laboratory for Physical Chemistry of Solid Surfaces, Collaborative Innovation Center of Chemistry for Energy Materials, and Department of Chemistry, College of Chemistry and Chemical Engineering, Xiamen University, Xiamen 361005, China. [2] Max Planck Institute for Polymer Research, Ackermannweg 10, 55128 Mainz, Germany. [3] Institute of Physical Chemistry, Johannes Gutenberg-Universitat Mainz, Duesbergweg 10-14, 55128 Mainz, Germany. [4] These authors contributed equally: Hao Hou, Xin-Jing Zhao, Chun Tang. ✉email: yuanzhi_tan@xmu.edu.cn

Cycloarenes are a class of polycyclic aromatic macrocycles composed of fully annulated benzene rings that enclose an inner cavity with C-H bonds pointing inward[1,2]. The synthesis of cycloarenes was initially part of an effort to investigate the nature of π-electron delocalization in aromatic systems[3–5]. Kekulene is the first successfully constructed (by Staab and Diederich[5] in 1978), and also the most studied, cycloarene. The experimental [1]H-nuclear magnetic resonance (NMR) spectrum of Kekulene shows a highly deshielded signal at 10.45 ppm[5] corresponding to the protons inside the inner cavity. This indicates that the π-electrons of Kekulene are delocalized within benzenoid rings (Clar model) rather than around the entire molecule (Kekule model)[5]. The same characteristics have later been derived for Kekulene homologs, including cyclo[*d,e,d,e,e,d,e,d,e,e*]decakisbenzene[6], septulene[7], octulene[8], and extended Kekulenes[9–12]. These cycloarenes provided valuable models to further develop theories on superaromaticity[13,14], global aromaticity[15], and related phenomena.

The inner cavity plays an important role in governing the molecular symmetry of cycloarenes[1,2]. Cycloarenes carrying a $C_2$-, $C_3$- or $C_6$-symmetric central cavity can be considered as molecular cutouts of defect-containing graphene. For instance, cyclo[*d,e,d,e,e,d,e,d,e,e*]decakisbenzene[6] represents a defective graphenic structure, which is formed by replacing two carbon atoms with four hydrogen atoms in the graphene lattice. Whereas, Kekulene and its π-extended analogs[9–12] are subunits of porous graphene[16] with six-member-ring pores. On the other hand, cycloarenes with a pentaradial, heptaradial, or octaradial inner cavity are non-graphitic and adopt a curved geometry. Notable members of this category include septulene[7] and octulene[8], which are hyperbolic nanocarbons with heptaradial and octaradial symmetry, respectively. Replacing the hexagonal central cavity in Kekulene with a smaller, pentagonal hole creates a fivefold-symmetric, bowl-shaped cycloarene, which can be named as quintulene (Fig. 1). Attempts to synthesize quintulene, launched by Staab and Sauer[17] in 1984, have remained unsuccessful, most probably due to the strong strain associated with its curved lattice.

Herein, we report the synthesis of extended quintulene (**1**), the still elusive member of the cycloarene family (Fig. 1). The construction of **1** was achieved through the π-extension of 5-cyclo-m-phenylene (**5CMP**). **1** is a fully benzenoid, bowl-shaped aromatic molecule and represents a naturally curved aromatic system complementing cylinder-shaped carbon nanobelts or nanohoops[18]. A remarkable feature of **1** is its tendency to form a metastable, but isolable dimer (**1**)$_2$ in solution. The structure of (**1**)$_2$ is characterized as a stacked bilayer carboncone complex. The kinetic and thermodynamic analysis discloses the dimerization process of **1** as an entropy-driven, second-order reaction with a substantial activation energy. In addition, the optical properties of **1** and (**1**)$_2$ are compared, revealing a H-type interlayer coupling in (**1**)$_2$.

## Results

**Synthesis of extended quintulene**. Inspired by the synthesis of extended Kekulenes[9], we selected **5CMP** with a pentagonal inner cavity as macrocyclic precursor (Fig. 2a). We first constructed the pentaradial polyphenylene **2** by coupling 2-bromo-5-mesityl-1,1'-biphenyl[19] with penta-borylated **5CMP** (**3**), which was obtained by Ir-catalyzed direct C−H borylation of **5CMP**[20] (Fig. 2a). The structure of **2** was confirmed by NMR spectroscopy and single crystal X-ray diffraction (Supplementary Figures 1–3 and Supplementary Note 3). As shown in Supplementary Figure 3, the macrocyclic ring of **2** adopts a nearly planar structure similar to that of **5CMP**[20]. It should be emphasized that the conversion of **2** to **1** by cyclodehydrogenation transforms a planar molecule into a curved one. Although oxidative cyclodehydrogenation of polyphenylene precursors have been widely used to generate planar polyaromatic hydrocarbons (PAHs), its application to the synthesis of bowl-shaped PAHs is more challenging owing to the unfavorable molecular strain[18,21,22].

After reaction optimization, complete cyclodehydrogenation of **2** to afford **1** was achieved by using 3.0 equiv of iron(III) chloride per hydrogen (3.0 equiv/H) at 0 °C for 10 h (Fig. 2a). The reaction mixture was purified first by silica gel flash column chromatography and then by high performance liquid chromatography (HPLC) using a 5PBB column. The component eluted at 73.5 min was collected (Supplementary Figure 4) and confirmed as **1** by matrix-assisted

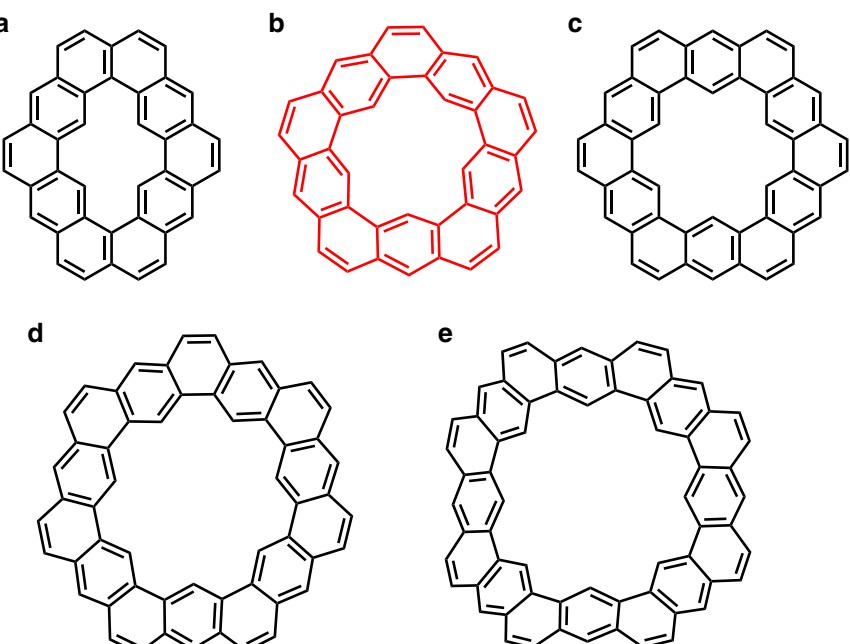

**Fig. 1 Structure of synthesized cycloarenes. a** cyclo[*d,e,d,e,e,d,e,d,e,e*]decakisbenzene, **b** quintulene, **c** Kekulene, **d** septulene, and **e** octulene. Quintulene was synthesized as its extended homolog in this report.

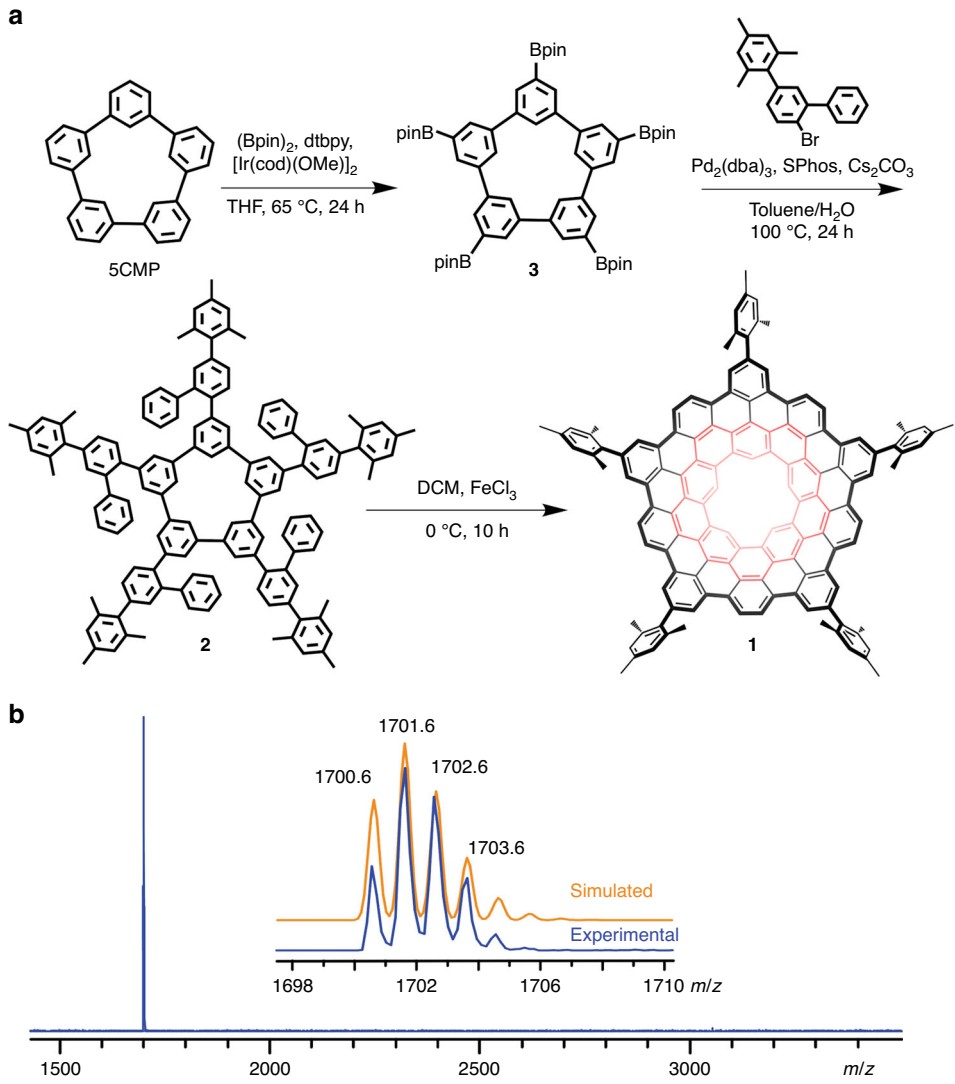

**Fig. 2 Synthetic route and mass spectra of 1. a** The synthetic route toward **1**. The quintulene unit in **1** is highlight in red. **b**. The mass spectrum of **1**. The isotopic distribution for the mass peak of **1** is shown as the insert figure. (Bpin)₂, bis(pinacolato)diboron; dtbpy, 4,4'-di-tert-butyl-2,2'-dipyridyl; DCM, dichloromethane; THF, tetrahydrofuran; [Ir(cod)(OMe)]₂, bis(1,5-cyclooctadiene)di-μ-methoxydiiridium(I); Pd₂(dba)₃, tris(dibenzylideneacetone) dipalladium(0); SPhos, 2-dicyclohexylphosphino-2',6'-dimethoxybiphenyl.

laser desorption/ionization-time of flight (MALDI-TOF) mass spectroscopy. A single molecular ion peak occurred at 1701.6 Da and the isotopic distribution precisely matched the theoretically calculated values according to the chemical formula $[C_{90}H_{25}(C_9H_{11})_5]$ (Fig. 2b). Moreover, no signal of dimer $(1)_2$ was observed when a freshly purified sample of **1** was analyzed (see below).

**Structural characterization of 1**. The $^1$H-NMR spectrum of **1** consisted of eight singlets with an intensity ratio of 1:2:2:1:1:3:3:3 (Fig. 3), in consistence with its $C_{5v}$ molecular symmetry. The hydrogens ($H_a$) inside the inner cavity appear as a singlet at 11.96 ppm (Fig. 3). Density functional theory (DFT) calculations (See Supplementary Note 4) demonstrate that **1** assumes a conical structure with a depth of 4.4 Å (Figs. 3a and 3b). Compared with the counterpart of **1** without the cavity, the pyramidalization angles[23] of carbon atoms in **1** are smaller (Supplementary Figure 5), which indicates a reduced curvature owing to the removal of the pentagonal ring at the center. Owing to the conical shape of **1** and the perpendicular orientation of the mesityl groups[19,24], the

ortho-methyl groups and the corresponding meta-aromatic hydrogens, respectively, are chemically inequivalent, as one of them points inward, and the other outward the cone (Fig. 3a, b). This is consistent with the presence of two signals for protons in phenyl ($H_e$, $H_g$: 7.13, 7.20 ppm, respectively) and three signals for methyl ($H_f$, $H_h$, $H_d$: 2.48, 2.32, 2.15 ppm, respectively). The calculated NMR spectrum based on the optimized structure of **1** largely matched the experimental one (Fig. 3c, Supplementary Figure 6 and Supplementary Note 4). Unlike the recently reported molecular carboncones[25,26], **1** shows a conical surface with zero Gaussian curvature and does not contain any bowl tip with positive-Gaussian curvature[18] (Fig. 3a, b). Therefore, **1** can be regarded as a naturally curved aromatic system aside from cylindrical carbon nanobelts and nanohoops[18].

To understand the π-electron structure of **1**, the nucleus-independent chemical shift (NICS) values of each rings and inner cavity in **1** at B3LYP/6-31 G(d,p) level were calculated. It appears that the individual hexagonal rings in **1** are either highly aromatic (ring I, III, IV: −8.5, −9.6, and −9.4 ppm, respectively) or nonaromatic (ring II, V: 0.6 and 2.1 ppm, respectively) (Fig. 3 and Supplementary Figure 7) and the distribution of aromatic rings

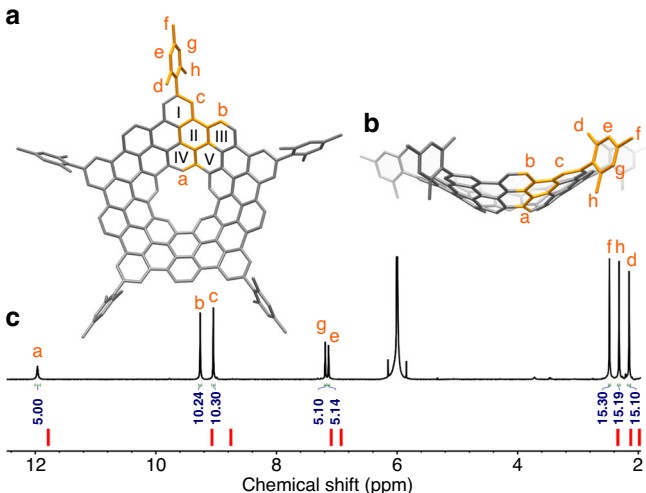

**Fig. 3 NMR characterization of 1. a** Top view of DFT-optimized **1**. The asymmetric unit is highlighted in orange. **b**. Side view of DFT-optimized **1**. All hydrogens in the asymmetric unit are labeled from a to h. The hexagonal rings of **1** are labeled by Roman numerals. **c**. $^1$H-NMR spectrum of **1** measured in $C_2Cl_4D_2$. The theoretical NMR spectrum of **1** is represented by the short vertical red lines below. Signal assignment is aided by 2D NMR spectroscopy (see Supplementary Note 1, Supplementary Figures 24–26).

fully obeys Clar's rule of isolated sextets. In addition, the anisotropy of induced current density (ACID) plot of **1** calculated at B3LYP/6-31 G(d,p) also shows an obvious diatropic ring current localized in those benzenoid rings (Supplementary Figure 7). Both NICS and ACID analysis clearly manifest that **1** adopts a localized π-electron structure and thus characterize **1** as a fully benzenoid bowl-shaped molecule. On the other hand, the inner cavity shows positive NICS value (5.1 ppm) and an overall paratropic ring current (Supplementary Figure 7), which explains the low-field resonance of $H_a$.

**Dimerization of 1**. When **1** was allowed to stand in solution, its $^1$H-NMR spectra displayed significant changes while the signals attributable to **1** showed a gradual decline, a new set of $^1$H-NMR peaks emerged (Fig. 4a and Supplementary Figures 8 and 9). From the time-dependent changes of the concentrations (Supplementary Figures 10–12), the chemical equilibration of dimerization was reached after 34.4 h at 30 °C. The equilibrated sample was subsequently analyzed by MALDI-TOF mass spectroscopy giving an additional peak at 3403.3 Da in addition to the signal of **1** (Supplementary Figure 13). The m/z value and isotopic distribution of the new peak precisely matched those of the dimer $(1)_2$ (Supplementary Figure 13). Therefore, the new set of $^1$H-NMR signals can be attributed to $(1)_2$. The slow dimerization even allowed us to obtain pure **1** and $(1)_2$ by HPLC separation (Fig. 4b), which represents a case of a metastable, but isolable dimer of a curved aromatic molecule. The MALDI-TOF spectrum of purified $(1)_2$ exhibited a single peak of dimer, without the signal of monomer (Fig. 4c). Similar to the molecular bilayer graphene that we reported earlier[19], the m/z peak intensity of $(1)_2$ decreased with increasing laser power (Supplementary Figure 14), providing additional support for the assumption of two stacked monomers.

The detailed structural validation of $(1)_2$ was achieved by NMR spectroscopy. Different from that of monomer **1**, the $^1$H-NMR spectrum of $(1)_2$ revealed two chemical shifts attributable to the inner protons, one at 12.74 ppm and another at 7.86 ppm, suggesting that the magnetic environments inside and outside the cone are not equivalent. In accordance with that, NICS

calculations along the quintuple axis of **1** indicate that the interior and exterior of the cone are magnetically shielded and deshielded, respectively (Supplementary Figure 15). We therefore propose that $(1)_2$ consists of two units arranged in a bilayer format via π−π stacking (Figs. 5a and 5b). The $^1$H-NMR spectrum of DFT-optimized $(1)_2$ displays the same number of proton signals and distribution pattern as the experimental NMR spectrum of $(1)_2$ (Fig. 5 and Supplementary Figure 16). The π−π stacked structure of $(1)_2$ could be further validated by the observation of 2D nuclear overhauser effect (NOE) signals attributable to interlayer proton coupling. The shortest distance between the two sets of inner protons ($H_a$, $H_a'$) on the two layers was found to be 3.6–3.8 Å and thus could enable spatial H···H coupling through space, which was clearly observed in NOE spectroscopy of $(1)_2$ (Supplementary Figure 17).

Assembly of aromatic molecules is often too fast to allow detailed kinetic measurement[27–30]. In contrast, the relatively slow dimerization of **1** (Fig. 3a) allowed us to measure relevant kinetic parameters by NMR spectroscopy. As shown in Fig. 6, the kinetic analysis clearly indicates second-order kinetics of the dimerization, with a surprisingly high activation energy of 74.3 ± 1.7 kJ mol$^{-1}$ for a π−π stacking complex. Subsequent analysis of the equilibrated sample in $C_2Cl_4D_2$ showed the binding constant ($K_a$) of $(1)_2$ ($3.6 \times 10^3$ at 30 °C) to be comparable to that of previously reported π−π dimers. The 1/T vs ln$K_a$ plot (Supplementary Figure 18) furnished an enthalpy change (ΔH) of 7.3 ± 1.0 kJ mol$^{-1}$ and an entropy change (ΔS) of 92.2 ± 3.1 J mol$^{-1}$ K$^{-1}$ based on the van't Hoff equation, manifesting an entropy-driven dimerization process. Then the solvent effects of dimerization was investigated in benzene ($C_6D_6$) (Supplementary Figures 19–23). The thermodynamics of dimerization of **1** in benzene revealed a ΔH of 1.2 ± 0.1 kJ mol$^{-1}$ and ΔS of 85.2 ± 0.3 J mol$^{-1}$ K$^{-1}$ (Supplementary Figure 23), which suggested that the dimerization was still entropy-driven but energetically more favorable in benzene. The kinetics of dimerization in benzene revealed a comparable activation energy of 80.2 ± 5.6 kJ mol$^{-1}$ (Supplementary Figure 22). The similar activation energies in benzene and tetrachloroethane could indicate an analogous transient state of dimerization in different solutions.

**Optical properties of 1 and $(1)_2$**. The slow dimerization also enables a reliable comparison of the optical properties of **1** and $(1)_2$. The absorption spectrum of **1** reveals a maximum absorption peak at 420 nm and additional bands at 445 and 479 nm, which represents the vibronic progression of the absorption bands caused by the coupling of the electronic transition to the C−C stretching of the annulated aromatic backbone (Supplementary Table 1).[31] The spectrum of $(1)_2$ exhibits a similar profile, but a blue-shifted absorption maximum and reduced low-energy absorption band when compared with **1**, which is the optical characteristic of H-type aggregation.[32] Geometrically, the stacking structure of $(1)_2$ is also in accordance with the "side-by-side" conformation of H-type aggregation.

The photoluminescence (PL) spectra of **1** and $(1)_2$ also shared similar features, including fine vibronic structures and a mirror-image relationship to the absorption spectra (Fig. 7a and Supplementary Table 2). The PL spectrum of $(1)_2$ demonstrated a small bathochromic shift (7–8 nm) over that of **1**. The small difference between the PL spectra of **1** and $(1)_2$ implies that the H-type coupling in $(1)_2$ is relatively weak[33], compared with that of planar analogs. The PL quantum yield of **1** was measured to be 11.7% in terms of the absolute method using integrating sphere. This is three times higher than those of recently reported carboncone molecules without inner cavity (quantum yield of PL 3.5%)[25,26], showing a defect-enhanced PL. In comparison, the PL

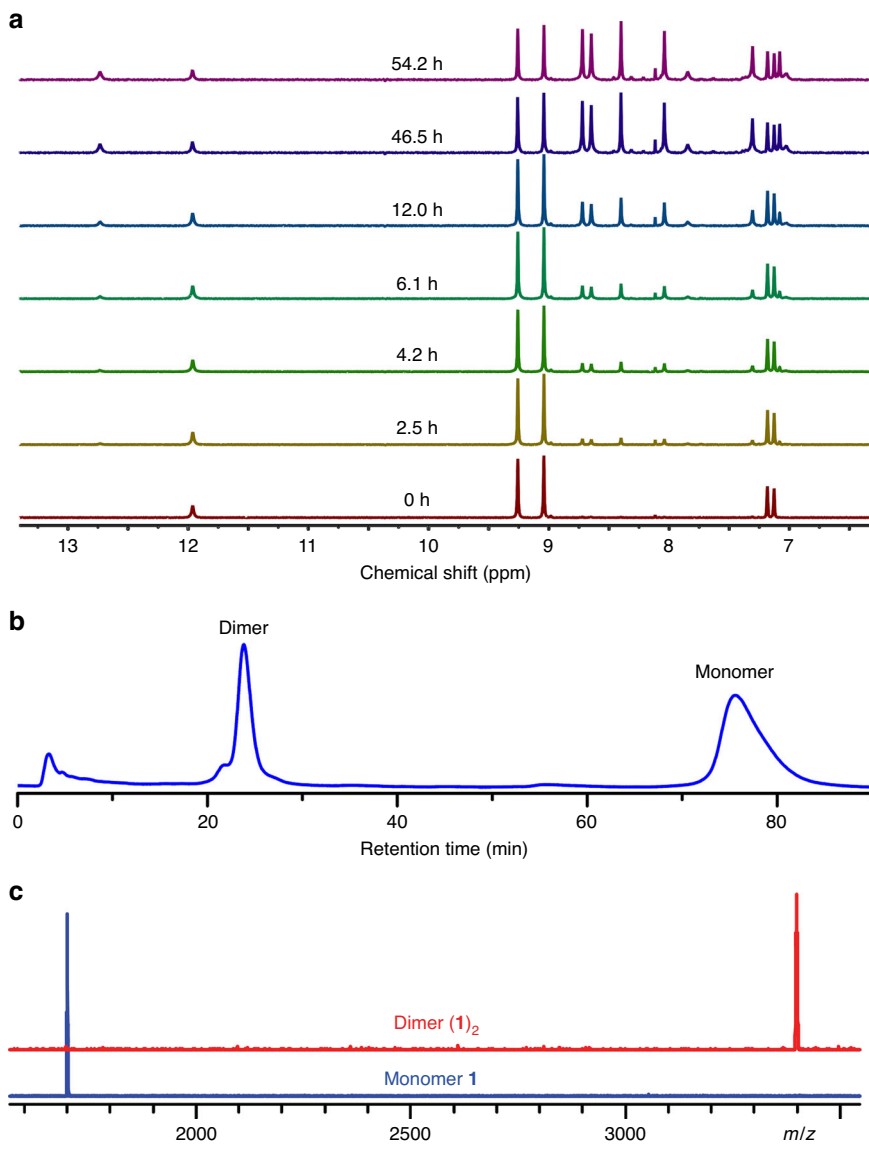

**Fig. 4 Dimerization of 1. a** Time-dependence of the $^1$H-NMR spectra of **1** at 30 °C measured in $C_2Cl_4D_2$. Chemical equilibrium is attained after 34.4 h. **b** HPLC chromatogram of the equilibrated sample using a 5PBB HPLC column (I.D. 10 × 250 mm) at a flow rate of 4 mL/min with toluene as mobile phase. **c** The mass spectra of **1** and (**1**)$_2$.

quantum yield of (**1**)$_2$ is reduced to 7.8%, probably owing to interlayer coupling. Time-resolved PL spectroscopy indicated that **1** and (**1**)$_2$ shared almost the same PL lifetime (14.4 and 14.3 ns, respectively) based on measurement at the maximum emission (Fig. 7b). Generally, the PL lifetime of excimers are distinctly longer[34], therefore the identical PL lifetime of **1** and (**1**)$_2$ suggests an absence of excimers during the excitation of (**1**)$_2$. The rate constants for the radiative ($k_r$) and nonradiative decay ($k_{nr}$) of **1** and (**1**)$_2$ were calculated from their PL lifetimes and quantum yields ($k_r = 8.1 \times 10^6$ s$^{-1}$ for **1** and $5.4 \times 10^6$ s$^{-1}$ for (**1**)$_2$, $k_{nr} = 6.1 \times 10^7$ s$^{-1}$ for **1** and $6.4 \times 10^7$ s$^{-1}$ for (**1**)$_2$). The obtained data manifested a suppressed $k_r$ of (**1**)$_2$, another optical signature of H-type aggregation[32], which confirms the H-type coupling in (**1**)$_2$.

## Discussion

Extended quintulene was synthesized through the π-extension of **5CMP** and unambiguously characterized by mass and NMR spectroscopy in combination with DFT calculations. The extended quintulene possesses a fivefold-symmetric conical structure, representing a naturally curved aromatic system. In solution, **1** undergoes dimerization to form (**1**)$_2$, which was characterized as a metastable π−π stacking bilayer complex. Kinetic and thermodynamic studies indicated that the dimerization of **1** in solution entails a high activation energy and is entropy-driven. A comparison of the optical properties of **1** and (**1**)$_2$ suggests a H-type coupling in (**1**)$_2$. Although PAHs are known to possess a high tendency toward aggregation in the solution, the present case is a unique example of a defined monomer-dimer equilibrium. The synthesis of extended quintulene **1** fills a long-standing gap in the cycloarene family. Also, it opens up many chemical opportunities including the further extension of the π-system and phase-forming derivatives of **1**. This would also suggest to study the semiconductor behavior of **1** together with the relevant charge-carrier transport as a function of the packing mode.

## Methods

**Synthesis of 2**. **3** (101 mg, 0.1 mmol), 2-bromo-5-mesitylbiphenyl (525 mg, 1.5 mmol), Pd$_2$(dba)$_3$ (46 mg, 0.05 mmol), SPhos (21 mg, 0.05 mmol) and

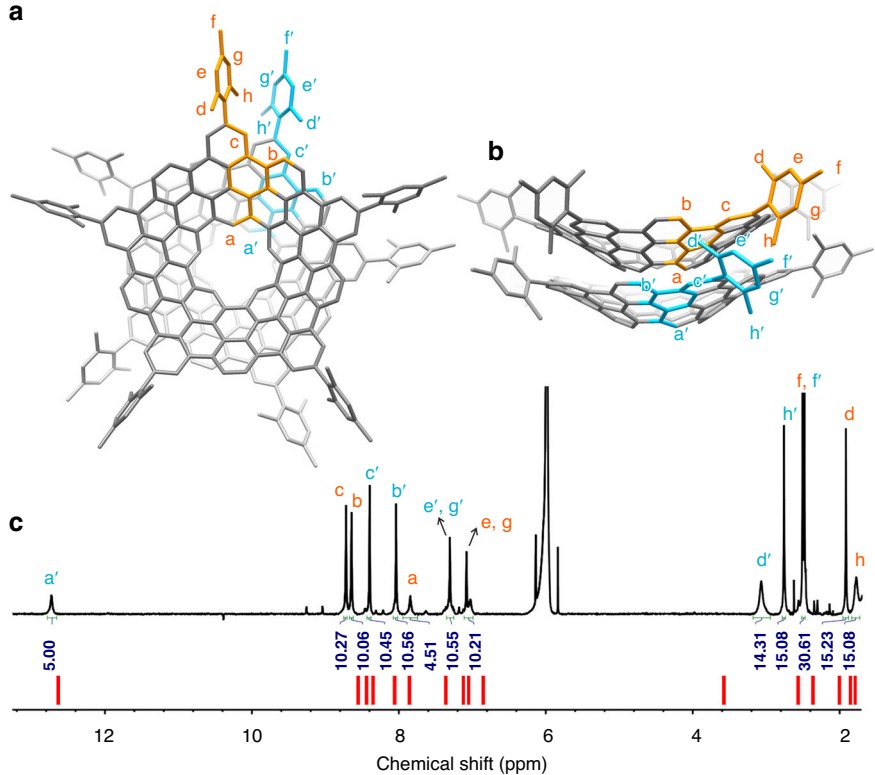

**Fig. 5 NMR characterization of (1)₂. a** The top view of the DFT-optimized structure of $(\mathbf{1})_2$. **b** The side view of the DFT-optimized structure of $(\mathbf{1})_2$. The asymmetric unit is highlighted in orange for one layer and blue for another layer. The hydrogen atoms in the asymmetric units are numbered. **c** $^1$H-NMR spectrum of $(\mathbf{1})_2$ measured in $C_2Cl_4D_2$. The calculated NMR spectrum of $(\mathbf{1})_2$ is represented as red lines. Signal assignment is aided by 2D NMR spectroscopy and DFT calculations (see Supplementary Note 2, Supplementary Figures 17 and 27–30).

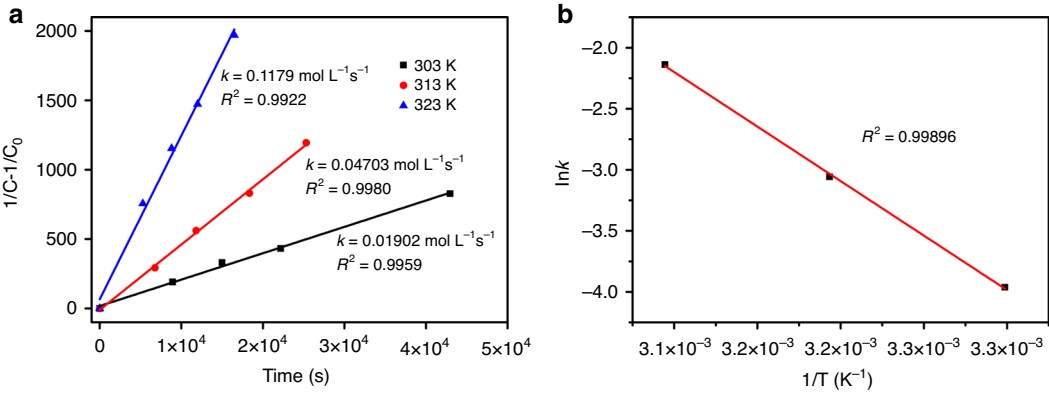

**Fig. 6 Kinetics of dimerization measured in $C_2Cl_4D_2$. a** Linear regression plot showing the second-order kinetics of the dimerization of **1**. **b** Arrhenius plot for the dimerization of **1**. ($C_0$, the initial concentration of monomer; C, temporal concentration of monomer; $k$, reaction rate constant; T, temperature).

$Cs_2CO_3$ (490 mg, 1.5 mmol) in toluene (3 mL) and water (1.5 mL) were stirred under argon at 100 °C for 24 h. The resulting mixture was cooled to room temperature, washed with water, and extracted with dichloromethane (DCM) (20 mL × 2). The organic layer was separated, dried with anhydrous $MgSO_4$, and then concentrated under reduced pressure. The obtained crude product of **2** was subsequently purified by silica gel column chromatography (DCM/petroleum ether, 1:3), affording a white solid (152 mg, 88%).

**Synthesis of 1**. A solution of **2** (20 mg, 0.012 mmol) in DCM (20 mL) was degassed by argon bubbling for 10 min at 0 °C, followed by the addition of iron (III) chloride (168 mg, 1.0 mmol). After stirring at 0 °C for 10 h with continuous argon bubbling, the reaction was quenched by methanol and the product was extracted by DCM (20 mL × 2). The organic layer was washed with water, dried with anhydrous $MgSO_4$ and concentrated. The crude product was obtained by silica gel flash column chromatography, filtered through a 0.2-µm organic membrane, and further purified using a 5PBB HPLC column (I.D. 10 × 250 mm) at a flow rate of 4 mL/min

with toluene as mobile phase. Pure product of **1** collected at the retention time of 73.5 min, followed by the immediate removal of the solvent in a rotary evaporator at room temperature. An amount of 1 mg of **1** was obtained with a calculated yield of 5%. It is worth noting that the low yield of **1** is owing to chlorination or incomplete ring closures during oxidative cyclodehydrogenation.

**Separation of (1)₂**. The equilibrated sample was separated using a 5PBB HPLC column (I.D. 10 × 250 mm) at a flow rate of 4 mL/min with toluene as mobile phase. Purified $(\mathbf{1})_2$ was obtained by collecting the eluted fraction at 23.8 min, followed, similarly as above, by the immediate removal of the solvent to avoid dissociation.

**Kinetics and thermodynamics of the dimerization**. The kinetics of dimerization was monitored and characterized by NMR spectroscopy. Specifically, the concentrations of **1** and $(\mathbf{1})_2$ were calculated by the relative intensities of their proton signals based upon the relation equation $2C_{dimer} + C_{monomer} = C_0$ ($C_0$, the initial

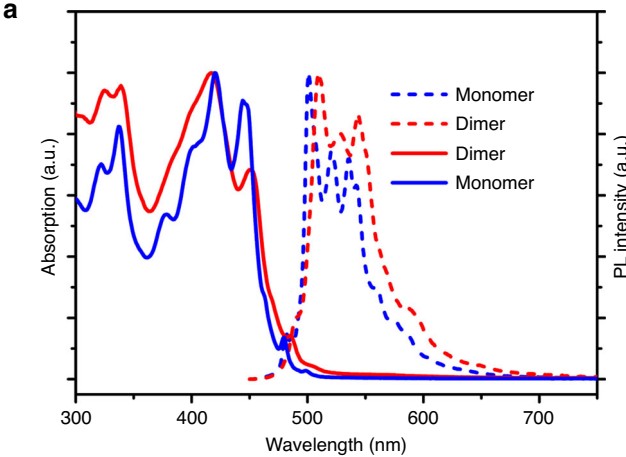

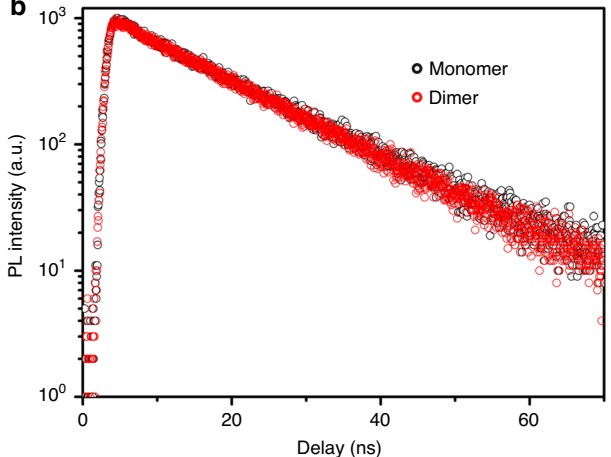

**Fig. 7 Optical properties of 1 and (1)₂. a** Absorption (solid lines) and photoluminescence (PL) (dashed lines) spectra of **1** (blue) and (**1**)₂ (red). **b**. Time-resolved PL kinetics of **1** (black) and (**1**)₂ (red).

concentration of monomer). The activation energy of the dimerization was evaluated by applying the Arrhenius equation on a series of reaction rate constants obtained at different temperatures. The concentrations of **1** and (**1**)₂ in the equilibrated mixture were measured following the same method as above.

## Data availability

The X-ray crystallographic coordinates for structure reported in this study have been deposited at the Cambridge Crystallographic Data Center (CCDC), under deposition number 1974252. These data can be obtained free of charge from The Cambridge Crystallographic Data Centre via www.ccdc.cam.ac.uk/data_request/cif.

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

## Acknowledgements

This work was financially supported by the Ministry of Science and Technology of China (2017YFA0204902, 2018YFA0209500) and the National Natural Science Foundation of China (21771155, 21721001).

## Author contributions

K.M. and Y.-Z.T. conceived and designed the project; H.H., X.-J.Z., Y.-Y.J., Z.-Y.D., and X.-R.W. conducted synthesis and completed the identification; C.T. conducted DFT calculations; H.H., L.-B.F. and D.-H.L. analyzed the NMR data; H.H., X.H., A.N., K.M., and Y.-Z.T. analyzed all the data and co-wrote the paper. All authors discussed the results and commented on the manuscript.

## Competing interests

The authors declare no competing interests.
