## [Peer Review File · Nature Communications]

REVIEWER COMMENTS

Reviewer #1 (Remarks to the Author):

The synthesis of extended quintulene by Tan et al. is a very nice piece of work, which can be highly recommended for publication in Nat. Comm. The crystal structure of the target is lacking, but the system has been properly characterized in solution. The observation of kinetically resolvable pi dimerization is a unique and highly interesting discovery. One point that is largely missing from the description is the solvent used for related investigations (kinetics, HPLC, etc.) This information should be provided by the authors. Also, if data in several solvents are available, they could be included (e.g. to show how the entropic factor is affected by the choice of solvent). Given the possibility of electronic communication between the two surfaces, it would be of interest to see the electrochemistry of 1 and (1)2. Can these systems be chemically oxidized to produce stable cationic states?

The quality of the manuscript is very high but it will benefit from one round of textual revision, to fix some unnecessary claims, missing references, and a few linguistic issues and typos.

Specific comments/issues

"ambiguously validated"

"isolatable"

"across the entire molecule"

"around" would be better here

"cyclo[d.e.d.e.e.d.e.d.e.e.]dekakisbenzene"

too many dots

"a graphenic structure with a di-vacancy defect¹⁵, which was formed by removing two carbon atoms from the graphene lattice"

This is not accurate, because the two C's are replaced with 4 H's rather than removed. It is OK to call it a defect, but it is not a vacancy in the proper sense and should be described differently

"C5-, C7- or C8-symmetric inner cavity"

This is a bit confusing. C5 initially seems to refer to the number of carbons in the ring rather than to molecular symmetry because of the missing italics. However, neither septulene nor octulene have complete C_n symmetry because of their hyperbolic distortion. The terms heptaradial/octaradial, though rare, seem to describe the structures better.

"the first fully benzenoid, bowl-shaped aromatic molecule"

The term "fully benzenoid" and even "benzenoid" is used inappropriately here. The first one has been introduced to describe those benzenoid systems in which all pi electrons can be partitioned into Clar sextets (e.g. HBC or triphenylene, cf. e.g. 10.1080/10406638.2014.918551, and 1st volume of Clar's PAH book). The least arbitrary definition of a benzenoid system requires that its carbon framework be a subgraph of graphene. In this sense, quintulene is no more benzenoid than corannulene. Overall, this is a rather confusing claim and it should be rephrased (several instances).

"a new class of naturally curved aromatic systems"

The molecule is a very novel structure, but I do not think it should be advertised as a "new class, etc."

As a fragment of a carbon nanocone it has precedents reported in refs. 25 and 26. It is not the first bowl-shaped coronoid either (e.g. 10.1002/anie.201208547, 10.1021/jacs.9b00683 should be cited to place this aspect in a literature context).

"electrocyclic ring closures"

electrocyclic is not appropriate here

"It should be noted, however, that the strain of the non-planar molecule is created at the very beginning, i.e. with the synthesis of 5CMP framework."

This is an interesting and relevant statement, but it is far from obvious whether the conversion of 2 into 1 leads to no further strain increase. The authors should provide a homodesmotic calculation to verify this.

"chemical inequivalent"

"two phenyl proton"

"Therefore, 1 can be regarded as a new class of naturally curved aromatic systems aside from cylindrical carbon nanobelts and nanorings¹⁸."

see the comment above

"Chemical equilibrium is attained after 34.4 h."

I am not sure a finite time can be assigned to the completion of an equilibration process. The authors should discuss figures S9-S11 in the manuscript, and name the quantity they determined using these plots.

"spatial H...H coupling"

Dipolar or through-space

"thermodynamic"

"The synthesis of quintulene 1 fills a long-standing gap in the cycloarene family"
extended quintulene

Level B alerts need to be explained in the cif file.

Reviewer #2 (Remarks to the Author):

The paper describes the synthesis and (mainly) self-aggregation property of a new hydrocarbon, coined extended quintulene, possessing a curved (bowl-shaped) structure with a hollow at the center. It may be regarded as a hybrid of a large benzenoid hydrocarbon with a cavity (reference 9) and large bowl-shaped molecules, carbon nanocone or carboncone, reported recently (references 25 and 26, respectively). Even though the synthesis was done in a clever design of the synthetic route starting from 5CMP, the individual method was established and has been used for the construction of large hydrocarbon frameworks like that of compound 1. The most important aspect of the paper therefore is the property of compound 1 as compared to those of the above-mentioned two types of hydrocarbons, i.e., flat benzenoid hydrocarbon with a hollow and cone-shaped solid molecules; in other words, the impact of the cavity on the physical property of 1. There indeed is one remarkable

issue, the slow dynamics of association in solution, which to my knowledge is the first example reported for large shape-persistent molecules like 1. Both thermodynamic and kinetic analyses of in-solution dimerization was performed, revealing that the dimerization is entropy-driven with a relatively large kinetic barrier. In view of this "hot" target molecule that would attract interest in diverse area of research ranging from organic chemistry to materials science, together with the interesting assembling property of the title compound, I recommend this paper for publication after revisions are made by addressing following comments.

1. What is the major driving force of (slow) aggregation of 1? And what is the role of the "hole" in the aggregation? Since the similar cone-shaped (solid) molecules are not reported to exhibit such property, there must be something to do with the cavity. Thermodynamic analysis shows it entropy-driven, strongly indicative of the role of desolvation. Is this something special in this system or nothing for a large aromatic hydrocarbon? In this respect, it is recommended to look into solvent effects if solubility permits. The solvent of the reported NMR measurement (not shown in the main text despite its potential importance!) should be included. The same is true for the kinetics, even though the solvent must be the same as that of thermodynamic measurement.
2. It is of course desirable to have X-ray structures for both monomer and dimer of 1, I suspect it not easy to obtain good crystals of either of them under the equilibrium condition. Nevertheless, it is possible in principle based on the Le Chatelier's law that the equilibrium can be shifted if one the components crystallizes out from the solution. Have the authors ever tried recrystallization at different (relatively higher than normal) temperatures for cooling?
3. Similarly, electrochemical properties may be determined under the equilibrium conditions, if the dissociation/association rates of charged species are slow enough for the potential scanning. Additionally, once electrochemical processes can be analyzed successfully, new dynamics may be revealed.
4. The most intriguing aspect of the NMR and theoretical ring current analysis is the paratropic nature of the inner cavity, surrounded by 15 sp² carbons. What does this paratropicity originate from?
5. It is tempting to ask about the effect of the cyclopentanoid at the center of the structure on the curvature of the molecular structure. This can be done by comparing the computational structures with the hypothetical molecule without the cavity.

Reviewers' comments:

Reviewer 1:

Reviewer #1 (Remarks to the Author):

The synthesis of extended quintulene by Tan et al. is a very nice piece of work, which can be highly recommended for publication in Nat. Comm. The crystal structure of the target is lacking, but the system has been properly characterized in solution. The observation of kinetically resolvable pi dimerization is a unique and highly interesting discovery.

One point that is largely missing from the description is the solvent used for related investigations (kinetics, HPLC, etc.) This information should be provided by the authors.

Response: Thanks for the comments. The information on solvent used for related investigations was provided in the revised manuscript and Supplementary Information.

Also, if data in several solvents are available, they could be included (e.g. to show how the entropic factor is affected by the choice of solvent).

Response: Thanks for the suggestion. We also monitored the dimerization by NMR spectroscopy in benzene (C_6D_6), in addition to the data in tetrachloroethane. The entropy changes upon dimerization in benzene and tetrachloroethane were quite close, calculated to be $85.2 \pm 0.3 \text{ J} \cdot \text{mol}^{-1} \text{K}^{-1}$ and $92.2 \pm 3.1 \text{ J} \cdot \text{mol}^{-1} \text{K}^{-1}$, respectively. The enthalpies of dimerization in benzene and tetrachloroethane were calculated as $1.2 \pm 0.1 \text{ kJ} \cdot \text{mol}^{-1}$ and $7.3 \pm 1.0 \text{ kJ} \cdot \text{mol}^{-1}$, respectively. This suggests that the dimerization was still entropy-driven but energetically more favorable in benzene. Similarly, the kinetics of dimerization in benzene was also investigated and calculated, revealing an activation energy of $80.2 \pm 5.6 \text{ kJ mol}^{-1}$, which is comparable with the activation energy measured in tetrachloroethane ($74.3 \pm 1.7 \text{ kJ mol}^{-1}$). The quite similar activation energies in benzene and tetrachloroethane could indicate an analogous transition state for dimerization in different solution.

The thermodynamic and kinetic data of dimerization in benzene were added in the revised manuscript and Supplementary Information.

Given the possibility of electronic communication between the two surfaces, it would be of interest to see the electrochemistry of **1** and (**1**)₂. Can these systems be chemically oxidized to produce stable cationic states?

Response: Thanks for the suggestion. We performed electrochemical measurements of **1** and (**1**)₂, however, unfortunately their cyclic voltammogram did not provide clear results due to the sparing solubility of **1** and (**1**)₂.

The quality of the manuscript is very high but it will benefit from one round of textual revision, to fix some unnecessary claims, missing references, and a few linguistic issues and typos.

Specific comments/issues

Comments: “ambiguously validated”

“isolatable”

“across the entire molecule”

“around” would be better here

Response: Thanks for the suggestions. These linguistic issues were corrected in the revised manuscript.

“cyclo[d.e.d.e.e.d.e.d.e.e.]dekakisbenzene”

too many dots

Response: Thanks for the suggestion. The nomenclature of cycloarene requests these dots, which cannot be omitted.

“a graphenic structure with a di-vacancy defect¹⁵, which was formed by removing two carbon atoms from the graphene lattice”

This is not accurate, because the two C’s are replaced with 4 H’s rather than removed. It is OK to call it a defect, but it is not a vacancy in the proper sense and should be described differently

Response: Thanks for the comment. The sentence was changed into “represents a defective graphenic structure, which is formed by replacing two carbon atoms with four hydrogen atoms in the graphene lattice”

“C5-, C7- or C8-symmetric inner cavity”

This is a bit confusing. C5 initially seems to refer to the number of carbons in the ring rather than to molecular symmetry because of the missing italics. However, neither septulene nor octulene have complete C_n symmetry because of their hyperbolic distortion. The terms heptaradial/octaradial, though rare, seem to describe the structures better.

Response: Thanks for the suggestion. “C5-, C7- or C8-symmetric inner cavity” was changed into “pentaradial, heptaradial or octaradial inner cavity”.

“the first fully benzenoid, bowl-shaped aromatic molecule”

The term “fully benzenoid” and even “benzenoid” is used inappropriately here. The first one has been introduced to describe those benzenoid systems in which all pi electrons can be partitioned into Clar sextets (e.g. HBC or triphenylene, cf. e.g. 10.1080/10406638.2014.918551, and 1st volume of Clar’s PAH book). The least arbitrary definition of a benzenoid system requires that its carbon framework be a subgraph of graphene. In this sense, quintulene is no more benzenoid than corannulene. Overall, this is a rather confusing claim and it should be rephrased

(several instances).

Response: Thanks for the valuable comments. Different from corannulene, all the π electrons of extended quintulene can be partitioned into Clar sextets (see the figure below). The nucleus-independent chemical shift (NICS) calculations also show that the hexagonal rings bearing Clar sextets are highly aromatic whereas the remaining hexagonal rings are nonaromatic. The aromatic sextets can classify extended quintulene as a fully benzenoid polycyclic aromatic hydrocarbon.

Figure 1 The Clar sextet valence bond structure of extended quintulene and corannulene. Clearly, all the π electrons of extended quintulene can be assigned to Clar sextets, while those of corannulene cannot.

Figure 2 NICS(0)_{iso} of extended quintulene.

The term “fully benzenoid” was also used to describe the electronic structure of carbon nanotubes (*J. Org. Chem.* **2004**, *69*, 4287-4291.).

The Clar sextet valence bond structure of extended quintulene was given in Figure S7 in the revised manuscript in.

“a new class of naturally curved aromatic systems”

The molecule is a very novel structure, but I do not think it should be advertised as a “new class, etc.” As a fragment of a carbon nanocone it has precedents reported in refs. 25 and 26. It is not the first bowl-shaped coronoid either (e.g. 10.1002/anie.201208547, 10.1021/jacs.9b00683 should be cited to place this aspect in

a literature context).

Response: Thanks for the comments. According to definition of coronoid in literature (*J. Chem. Inf. Comput. Sci.* **1990**, *30*, 210-222.) “A coronoid (system) is a polyhex with a (corona) hole of the size of more than one hexagon.”, the compounds reported in 10.1002/anie.201208547 and 10.1021/jacs.9b00683 contain the pentagons in the conjugated π system, therefore they cannot be classified as coronoids. These compounds in 10.1002/anie.201208547 and 10.1021/jacs.9b00683 do not satisfy the definition of cycloarenes “polycyclic aromatic compounds in which, by a combination of angular and linear annulations of benzene units, fully annelated macrocyclic systems are present enclosing a cavity into which carbon–hydrogen bonds point” (*Chem. Soc. Rev.* **2017**, *46*, 7-20.), due to the pentagonal rings in their skeleton. However, we agree with the reviewer that it is better not to advertise the compound **1** as a “new class, etc.” and we removed the “new class” in the revised manuscript.

“electrocyclic ring closures”

electrocyclic is not appropriate here

Response: Thanks for the suggestion. “Although electrocyclic ring closures based upon oxidative cyclodehydrogenation” was changed into “Although oxidative cyclodehydrogenation”.

“It should be noted, however, that the strain of the non-planar molecule is created at the very beginning, i.e. with the synthesis of 5CMP framework.”

This is an interesting and relevant statement, but it is far from obvious whether the conversion of **2** into **1** leads to no further strain increase. The authors should provide a homodesmotic calculation to verify this.

Response: Thanks for the comments. The strain of precursor and extended quintulene was calculated to be $23.4 \text{ kcal}\cdot\text{mol}^{-1}$ and $49.5 \text{ kcal}\cdot\text{mol}^{-1}$ by theoretical homodesmotic reactions, respectively. Although the precursor bears strain already, the conversion of the precursor into extended quintulene leads to additional strain. Thus, the sentence “It should be noted, however, that the strain of the non-planar molecule is created at the very beginning, i.e. with the synthesis of 5CMP framework.” was removed in the revised manuscript.

Figure 3. Theoretical homodesmotic reaction from the macrocyclic precursor **2** to

strain-free molecular subunits.

Figure 4. Theoretical homodesmotic reaction from the extended quintulene to strain-free molecular subunits.

“chemical inequivalent”

“two phenyl proton”

Response: Thanks for the comments. These typos were corrected.

“Therefore, 1 can be regarded as a new class of naturally curved aromatic systems aside from cylindrical carbon nanobelts and nanorings¹⁸.”

see the comment above

Response: Thanks for the comments. The statement of “new class” was removed.

“Chemical equilibrium is attained after 34.4 h.”

I am not sure a finite time can be assigned to the completion of an equilibration process. The authors should discuss figures S9-S11 in the manuscript, and name the quantity they determined using these plots.

Response: Thanks for the comments. We agree that direct measurement of equilibration time is difficult. It can be theoretically calculated from the kinetic plot, such as Figure S9-S11. The related discussion was added in the Figure caption.

Figure S10 The plot of $(1/C-1/C_0)$ vs time at 30 °C measured in $C_2Cl_4D_2$. (C_0 , the

initial concentration of monomer; C , time-dependent concentration of monomer). Before equilibration, the time-dependent concentration of monomer (C) decreases with increasing reaction time. According to second-order kinetics, $1/C - 1/C_0$ should increase linearly with time (red line) before equilibration. The experimental data fit the red line well, which validates the second-order kinetics for the assembly of extended quintulene. After equilibration, C remains unchanged, thus $1/C - 1/C_0$ vs time should be a line parallel to the time axis (blue line). Then the intersection of the red and blue lines is the point when chemical equilibrium is attained (34.4 h).

“spatial H···H coupling”

Dipolar or through-space

Response: Thanks for the comments. The spatial H···H coupling is through space. “thus could enable spatial H···H coupling” was changed into “thus could enable H···H coupling through space”

“thermodynamic”

Response: The typo was corrected.

“The synthesis of quintulene 1 fills a long-standing gap in the cycloarene family”
extended quintulene

Response: “The synthesis of quintulene” was corrected into “The synthesis of extended quintulene”

Level B alerts need to be explained in the cif file.

Response: There are two Level B alerts.

“Ratio Observed / Unique Reflections (too) Low .. 38% Check

Low Bond Precision on C-C Bonds 0.01235 Ang.”

The crystal of precursor **2** easily effloresces due to cocrystallized solvent molecules. The efflorescence decreases the quality of crystal, which leads the low ratio of observed / unique reflections and consequent low bond precision.

The response to Level B alerts was added in the cif file.

Reviewer #2 (Remarks to the Author):

The paper describes the synthesis and (mainly) self-aggregation property of a new hydrocarbon, coined extended quifulene, possessing a curved (bowl-shaped) structure with a hollow at the center. It may be regarded as a hybrid of a large benzenoid hydrocarbon with a cavity (reference 9) and large bowl-shaped molecules, carbon nanocone or carboncone, reported recently (references 25 and 26, respectively). Even though the synthesis was done in a clever design of the synthetic route starting from 5CMP, the individual method was established and has been used for the construction of large hydrocarbon frameworks like that of compound 1. The most important aspect of the paper therefore is the property of compound 1 as compared to those of the above-mentioned two types of hydrocarbons, i.e., flat benzenoid hydrocarbon with a hollow and cone-shaped solid molecules; in other words, the impact of the cavity on the physical property of 1. There indeed is one remarkable issue, the slow dynamics of association in solution, which to my knowledge is the first example reported for large shape-persistent molecules like 1. Both thermodynamic and kinetic analyses of in-solution dimerization was performed, revealing that the dimerization is entropy-driven with a relatively large kinetic barrier. In view of this “hot” target molecule that would attract interest in diverse area of research ranging from organic chemistry to materials science, together with the interesting assembling property of the title compound, I recommend this paper for publication after revisions are made by addressing following comments.

1. What is the major driving force of (slow) aggregation of 1? And what is the role of the “hole” in the aggregation? Since the similar cone-shaped (solid) molecules are not reported to exhibit such property, there must be something to do with the cavity. Thermodynamic analysis shows it entropy-driven, strongly indicative of the role of desolvation. Is this something special in this system or nothing for a large aromatic hydrocarbon? In this respect, it is recommended to look into solvent effects if solubility permits. The solvent of the reported NMR measurement (not shown in the main text despite its potential importance!) should be included. The same is true for the kinetics, even though the solvent must be the same as that of thermodynamic measurement.

Response: Thank you very much for the comments.

First, the major driving force of dimerization is the π - π interaction. Generally, large planar polycyclic aromatic hydrocarbons (PAHs) can form a stacked columnar supramolecular structure, the stability of which increases with the size of PAHs. The formation of discrete dimers requires the balance of the π - π stacking of conjugated π systems with the effects of steric hindrance effects of peripheral groups. The peripheral groups have been found to be effective for the modulation of aggregation behavior of PAHs. For example, bulky mesityl group have been introduced at the periphery to hinder interlayer stacking: hexabenzocoronene (HBC) functionalized with mesityl groups exists as monomer (*Org. Lett.* **2012**, *14*, 2472-2475.). If the inner core of PAH increases, the enhanced π - π interaction can prompt the PAHs decorated with bulky groups to aggregate again. Recently, our group reported three cases of

stable and discrete dimers of planar PAHs molecules, which contain nano-sized π systems and peripheral mesityl groups (*Nat. Commun.* **2019**, *10*, 3057. and *Sci. Adv.* **2020**, *6*, eaay8541.) (Figure 5). The carbon nanocone molecules reported in refs. 25 and 26 were functionalized by mesityl or similar 2,6-dimethylphenyl groups, but their π conjugated core is smaller than that of **1** (Figure 6) and thus might not be big enough to suppress the steric hindrance induced by bulky peripheral groups. Therefore, they do not show aggregation in solution. In this view, the synthesis of PAHs with extended π systems will be decisive for the self-assembly.

[Redacted]

Figure 5. Mesityl functionalized PAHs.

Figure 6. Inner conjugated cores for **1** (C_{90}) and carbon nanocone (C_{70}) reported in refs 25 and 26.

The dynamic aggregation of **1** results from its curved structure. Compared with the planar analogs, the interlayer π - π stacking of curved systems could be weaker. Thus, **1** forms a metastable dimer instead of a stable dimer in the cases of the planar analogs (*Nat. Commun.* **2019**, *10*, 3057. and *Sci. Adv.* **2020**, *6*, eaay8541.).

Two discrete dimers of PAHs which we reported - one with the inner cavity (C_{108})₂ and the other without the inner cavity (C_{114})₂ (Figure 5) - showed the same assembly behavior and formed stable dimers. Thus we assume that the effects of the inner

cavity on dimerization of **1** will be small.

Second, we agree that the desolvation is important for dimerization. The solvent used in NMR measurements for thermodynamic and kinetic investigations was the same and specified in the revised manuscript. To evaluate the solvent effects of dimerization, we carried out NMR measurements of the dimerization in benzene (C_6D_6), in addition to the data in tetrachloroethane. The entropy changes of dimerization in benzene and tetrachloroethane were quite close, calculated to be $85.2 \pm 0.3 \text{ J} \cdot \text{mol}^{-1} \text{K}^{-1}$ and $92.2 \pm 3.1 \text{ J} \cdot \text{mol}^{-1} \text{K}^{-1}$, respectively. The enthalpies of dimerization in benzene and tetrachloroethane were calculated to be $1.2 \pm 0.1 \text{ kJ} \cdot \text{mol}^{-1}$ and $7.3 \pm 1.0 \text{ kJ} \cdot \text{mol}^{-1}$, respectively. This suggests that the dimerization was still entropy-driven but energetically more favorable in benzene. Similarly, the kinetics of dimerization in benzene was also investigated and calculated, revealing an activation energy of $80.2 \pm 5.6 \text{ kJ mol}^{-1}$, which is comparable to the activation energy measured in tetrachloroethane ($74.3 \pm 1.7 \text{ kJ mol}^{-1}$). The comparable activation energies in benzene and tetrachloroethane could indicate a similar transient state of dimerization in different solutions.

The thermodynamic and kinetic data of dimerization in benzene were added in the revised manuscript and Supplementary information.

2. It is of course desirable to have X-ray structures for both monomer and dimer of **1**, I suspect it not easy to obtain good crystals of either of them under the equilibrium condition. Nevertheless, it is possible in principle based on the Le Chatelier's law that the equilibrium can be shifted if one the components crystallizes out from the solution. Have the authors ever tried recrystallization at different (relatively higher than normal) temperatures for cooling?

Response: Thanks for the suggestion. Indeed, we have tried to crystallize **1** for more than one year since we completed the synthesis. A series of methods for crystallization, including solvent evaporation, solvent diffusion, and programed cooling, were tried. But unfortunately all the attempts failed. There may be different reasons for this failure:

In principle, according to Le Chatelier's law, the equilibrium will be shifted during crystallization. However, the kinetics of the dimerization of **1** is slow and its timescale is close to that of crystallization. Thus they behave like a mixture of two compounds and both monomer and dimer precipitate out from solution at the same time, which leads to the formation of an amorphous solid instead of crystals.

On the other hand, in the solid state, the dimer and monomer are stable, but in solution they will transform into monomer or dimer and afford a mixture again during the crystallization process, which generally requires days.

3. Similarly, electrochemical properties may be determined under the equilibrium conditions, if the dissociation/association rates of charged species are slow enough for the potential scanning. Additionally, once electrochemical processes can be analyzed successfully, new dynamics may be revealed.

Response: Thanks for the suggestion. We performed the electrochemical measurement of **1** and $(\mathbf{1})_2$, however unfortunately their cyclic voltammogram did not

give clear results due to the sparing solubility of **1** and (**1**)₂.

4. The most intriguing aspect of the NMR and theoretical ring current analysis is the paratropic nature of the inner cavity, surrounded by 15 sp² carbons. What does this paratropicity originate from?

Response: Thanks for the comment. Compound **1** adopts a fully benzenoid electronic structure and all the π electrons can be represented by Clar sextets. The rings bearing Clar sextets exhibit strong diatropic ring currents (blue cycles in Figure 7). Then the inner cavity forms a paratropic ring (red cycle) current by connecting the strong diatropic ring currents (blue cycles) of aromatic sextets around the inner cavity (Figure 7).

Figure 7 Anisotropy of the induced current density (AICD) plot and NICS(0)_{iso} of **1**. a. the anisotropy of the induced current density (AICD) plot of **1**. Blue and red cycles represent diatropic and paratropic currents, respectively. The magnetic field is perpendicular to the mean plane of the molecule and points downwards. b. The NICS(0)_{iso} values of **1** are shown at the corresponding positions. All the π electrons of **1** can be partitioned into Clar sextets. The positive and negative values are labelled in red and blue, respectively.

5. It is tempting to ask about the effect of the cyclopentanoid at the center of the structure on the curvature of the molecular structure. This can be done by comparing the computational structures with the hypothetical molecule without the cavity.

Response: Thanks for the suggestions. We calculated the counterpart of **1** without the cavity by DFT calculations. Then we used the pyramidalization angle (*Science* **1993**, *261*, 1545-1550.) to evaluate the curvature of these molecules. As shown in the Figure S5, the counterpart without inner cavity shows higher curvature of the carbon skeleton, especially for the carbon atoms close to the pentagonal ring at the center. The related discussion and Figure S5 was added in the revised manuscript.

Figure S5 The pyramidalization angle of the carbon atoms in **1** and its counterpart without the cavity. The structures were optimized by DFT calculations.

REVIEWERS' COMMENTS:

Reviewer #1 (Remarks to the Author):

The authors have provided an extensive revision and rebuttal to my comments, and I am mostly satisfied with the changes and explanations. I believe the manuscript may be recommended for publication once the following two minor issues are addressed.

The usage of dots in cycloarene names is a minor issue but it deserves further attention from the authors for the benefit of future publications on this topic. I believe that the final dot should not be used. Apparently the confusion starts with a prominent typo in the title of 10.1002/anie.198607421, which was reiterated in a modern review 10.1039/C6CS00174B. Staab's nomenclature was formally introduced in 10.1002/cber.19831161021, where the final dot is never used. The latter convention (with dots replaced with commas) was adopted in 10.1039/C5CS00185D.

The homodesmotic reaction used by authors in Figure 4 (in the rebuttal) is not the best choice, because the authors cut the molecule across a Clar sextet while using naphthalene as the reference system. Naphthalene is not "fully benzenoid" and the differences in pi conjugation may be expected to show up in the calculated energy. I would suggest the authors use a homodesmotic calculation analogous to the method used in 10.1002/anie.201203266.

Reviewer #2 (Remarks to the Author):

The paper was improved by addressing satisfactorily the comments/suggestions and is now recommended for publication. Please correct the still remaining error however.
page 1, line 17, unambiguously

Reviewers' comments:

Reviewer #1 (Remarks to the Author):

The authors have provided an extensive revision and rebuttal to my comments, and I am mostly satisfied with the changes and explanations. I believe the manuscript may be recommended for publication once the following two minor issues are addressed.

The usage of dots in cycloarene names is a minor issue but it deserves further attention from the authors for the benefit of future publications on this topic. I believe that the final dot should not be used. Apparently the confusion starts with a prominent typo in the title of 10.1002/anie.198607421, which was reiterated in a modern review 10.1039/C6CS00174B. Staab's nomenclature was formally introduced in 10.1002/cber.19831161021, where the final dot is never used. The latter convention (with dots replaced with commas) was adopted in 10.1039/C5CS00185D.

Response: Thanks for the suggestion. The nomenclature of cycloarene was corrected according to the suggestion of reviewer.

The homodesmotic reaction used by authors in Figure 4 (in the rebuttal) is not the best choice, because the authors cut the molecule across a Clar sextet while using naphthalene as the reference system. Naphthalene is not "fully benzenoid" and the differences in π conjugation may be expected to show up in the calculated energy. I would suggest the authors use a homodesmotic calculation analogous to the method used in 10.1002/anie.201203266.

Response: Thanks for the suggestion. The homodesmotic reactions following the reviewer's suggestion were calculated (Figure 1). We agree with reviewer that the difference in π conjugation between reactant and product in the homodesmotic reaction will show up in the calculated energy. As shown in Figure 1, the final fully-cyclodehydrogenated **1** had the lowest energy, which could be attributed to the large energy gain from the extension of π -conjugation. We think the homodesmotic reactions are not suitable for the strain calculation of molecules with fully-fused π -system, owing to unavoidable change of π conjugation in homodesmotic reaction. The strain calculation of **1** was not included in the manuscript.

Figure 1. The development of relative energy in the synthesis of **1**.

Reviewer #2 (Remarks to the Author):

The paper was improved by addressing satisfactorily the comments/suggestions and is now recommended for publication. Please correct the still remaining error however. page 1, line17, unambiguously

Response: Thanks for the correction. It was corrected in the revision.